# The Impact of Immune System Aging on Infectious Diseases

**DOI:** 10.3390/microorganisms12040775

**Published:** 2024-04-11

**Authors:** Eugenia Quiros-Roldan, Alessandra Sottini, Pier Giorgio Natali, Luisa Imberti

**Affiliations:** 1Department of Infectious and Tropical Diseases, ASST- Spedali Civili and DSCS- University of Brescia, 25123 Brescia, Italy; maria.quirosroldan@unibs.it; 2Clinical Chemistry Laboratory, Services Department, ASST Spedali Civili of Brescia, 25123 Brescia, Italy; alessandra.sottini@asst-spedalicivili.it; 3Mediterranean Task Force for Cancer Control (MTCC), Via Pizzo Bernina, 14, 00141 Rome, Italy; natalipg2002@yahoo.it; 4Section of Microbiology, University of Brescia, P. le Spedali Civili, 1, 25123 Brescia, Italy

**Keywords:** aging, infectious disease, immunosenescence, inflammaging, vaccination

## Abstract

Immune system aging is becoming a field of increasing public health interest because of prolonged life expectancy, which is not paralleled by an increase in health expectancy. As age progresses, innate and adaptive immune systems undergo changes, which are defined, respectively, as inflammaging and immune senescence. A wealth of available data demonstrates that these two conditions are closely linked, leading to a greater vulnerability of elderly subjects to viral, bacterial, and opportunistic infections as well as lower post-vaccination protection. To face this novel scenario, an in-depth assessment of the immune players involved in this changing epidemiology is demanded regarding the individual and concerted involvement of immune cells and mediators within endogenous and exogenous factors and co-morbidities. This review provides an overall updated description of the changes affecting the aging immune system, which may be of help in understanding the underlying mechanisms associated with the main age-associated infectious diseases.

## 1. Introduction

Aging is not only a gradual and irreversible pathophysiological process, but also a complex phenomenon from a social, psychological, and biological point of view. Since it can impact the functions of many organs and systems, it represents a major risk factor for the onset and/or progression of a wide spectrum of aging-related disorders [1]. As life expectancy is predicted to increase, understanding the mechanisms leading to immune senescence is of major public health relevance, fostering the development of more focused preventive actions, as well as the development of new preventive and therapeutic interventions.

Aging has gone through evolving definitions over the years, such as “a persistent decline in the age-specific fitness components of an organism due to internal physiological deterioration” [2] or “a progressive loss of function accompanied by decreasing fertility and increasing mortality with advancing age” [3]. More recently, a consensus has been reached on framing aging as a “normal way of functioning in biology from a certain age onwards” [4]. For instance, aging in different organisms (especially mammals) has been initially proposed to encompass nine common features, including genomic instability, telomere shortening attrition, epigenetic alteration, deregulation of nutrient sensing, mitochondrial dysfunction, cellular senescence, stem cell exhaustion, alteration of intercellular communication, and loss of proteostasis [5]. Over the years, the finding that diminished proteostasis is different from compromised autophagy has enabled the inclusion of the latter in the hallmarks of aging [6]. In addition, three other distinctive derangements, such as alteration of metabolic pathways, adaptation to stress, and, more importantly, chronic inflammation, have been added to the list [7]. These hallmarks, which are strongly correlated with each other, satisfy the following three premises: 1. they are aging-related; 2. the possibility of experimentally accelerating aging by acting on these markers; and 3. the opportunity to decelerate, stop or reverse aging with a combination of preventive and therapeutic interventions.

In humans, senescent cells of various lineages accumulate at various body sites at different rates, from 2- to 20-fold, when comparing young (<35 years) to old (>65 years) healthy individuals [8]. Although all cell types can undergo senescence during aging, this process mainly affects fibroblasts, endothelial cells, chondrocytes, tenocytes, skin and immune cells [9,10,11]. The immune system is one of the most ubiquitous systems of the organism, which can protect the human body from internal or external agents and interacts with neural, circulatory and other systems. Therefore, its alteration may result in increased incidence of many age-related diseases.

## 2. Aging of Immune System Components in Physiological Conditions

Age-related immune system changes are grouped under the term “immunosenescence”, used over the years to define “the state of dysregulated immune function that contributes to the increased susceptibility of the elderly to infections and, possibly, autoimmune diseases and cancer” [12]; “the changes in the immune system associated with age” [13]; and “a gradual and subjective decline of the immune system and host defense mechanisms” [14]. More generally, immunosenescence refers to a series of complex changes leading to altered innate and adaptive immune system functions which, overall, may result in a state of immunodeficiency [15]. Immunosenescence has been considered a harmful event for many years, due to its involvement in the progressive reduction in the ability to trigger effective antibody and cellular responses against infections and vaccinations. It has also been considered a negative process because it leads to a type of inflammation called “inflammaging”, which defines a reduction in the capability to cope with a variety of stressors and a progressive increase in the pro-inflammatory status [16]. Indeed, inflammaging is characterized by an unbalanced increase in systemic pro-inflammatory cytokines, such as interleukin (IL)-1, IL-6, and tumor necrosis factor (TNF)-α, and reduced levels of anti-inflammatory cytokines, such as IL-10, transforming growth factor-β [17]. Therefore, inflammaging is a chronic, asymptomatic, sterile, low-grade inflammation that occurs in old age in the absence of signs of infection. This chronic inflamed state was initially described as a harmful event with profound adverse health effects that contributed to biological aging and the development of age-related pathologies. In particular, senescent immune cells, by producing cytokines, chemokines, growth factors, proteases, and angiogenic factors, acquire the so-called “senescence-associated secretory phenotype” (SASP) [18]. The persistent and non-resolved production of pro-inflammatory mediators, concomitantly with the adoption of lifestyle factors, including smoking, obesity, alcohol, lack of exercise, and exposure to ultraviolet radiation, are known to increase the risk for age-related multi-organ/system diseases (Figure 1) [15,19,20,21,22] and premature morbidity and mortality.

In recent years, the perception regarding immunosenescence and inflammaging events has changed considerably. Indeed, while the accumulation of pro-inflammatory factors and inflammaging were suggested as standing at the origin of most diseases of the elderly, age-related inflammation showed a closer correlation with longevity than any other parameters [22,23]. In addition, age-related thymic involution (see below), which leads to a restriction of the T-cell receptor (TCR) repertoire, may also result in lower energy consumption, favoring other body survival-supportive functions and activities [23]. Therefore, it has been proposed that, without the existence of the immunosenescence/inflammation binomial, which represents two aspects of the same phenomenon, human longevity would be greatly reduced. From an evolutionary point of view, immunosenescence is an optimization of the resources of the aging body, even if it ultimately may lead to pathologies and death [23]. An alternative possibility is that immunosenescence is the phenomenon in which adaptive immunity decreases over time, while inflammaging is a phenomenon in which innate immunity is activated [24].

According to this theory, inflammaging and immunosenescence progress in parallel, sustaining a mutually maintained vicious loop: the increased release of inflammatory mediators induced by inflammaging contributes to inhibiting the adaptive immune system and promoting immunosenescence; vice versa, the decrease in the adaptive immune system response strengthens the stimulation of the innate immune system (in an attempt to protect the organism) and maintains inflammaging. One of the most recent hypotheses on the meaning of immune system age-related changes is that these modifications are linked to the immune system response to permanent stress. This novel vision considers these changes as a permanent process of adaptation to stress that could have either beneficial or harmful consequences, depending on both genetic and environmental exposure. The prevalence of one of the conditions may result either in healthy longevity or pathological aging burdened by age-related diseases [25].

Overall, aging of the immune system in physiological conditions involves complex alterations in both innate and adaptive immunity, leading to decreased immune responsiveness, increased susceptibility to harmful agents and infections, and impaired vaccine efficacy. Understanding these changes is essential for developing strategies to promote healthy aging and improve immune function in older adults.

### 2.1. Age-Associated Changes in the Immune Compartments

Several age-related modifications affect the innate and adaptive immune system [26,27]. In aging bone marrow, hematopoietic stem cells show reduced self-renewal potential and increased skewing toward myelopoiesis [28], probably because of compromised Pax5 expression and Rag2 recombinase function and bone marrow niche alterations [29]. The number of circulating neutrophils does not significantly change [30], although the function of these cells may be compromised in aging, since activated neutrophils are more prone to apoptosis and deficient in phagocytosis and chemotaxis [29]. In addition, aging, by hampering neutrophil extracellular traps (NETs) efficacy, reduces NETosis, the neutrophil-mediated defense mechanism in which DNA and enzymes are extruded, forming a network trapping and killing different pathogens [31].

Aged tissue-derived macrophages may display compromised phagocytosis and an altered response to lipopolysaccharide [32]. Macrophages, upon stimulation, may undergo pyroptosis, also known as cellular inflammatory necrosis, which is different from other kinds of cell death. Pyroptosis is thought to play a key role in the clearance of infectious agents by exposing pathogenic antigens to the adaptive immune system and secreting cytokines and eicosanoids to promote inflammatory and repair responses [33]. More recently, the role of pyroptosis in the age-related diseases has attracted increased research attention. However, the elucidation of pyroptosis pathophysiology is still a work in progress, and it is not known whether modifications of a specific pyroptotic pathway may be beneficial, or may upregulate other pathways implicated in age-associated diseases that may paradoxically exacerbate disease progression [34].

Although information regarding the numbers and performance of dendritic cells (DCs) is relatively scarce, the number of circulating conventional and plasmacytoid DCs appears to be reduced in frail, healthy elderly [35]. While natural killer (NK) cells significantly increase, with the majority of these cells being CD56dim, their proliferative ability declines with age [36].

One of the main and early investigated adaptive immune system impairments occurs as a result of thymic involution, such as loss of organ mass, disruption of architecture, enhancement of perivascular space, and increased adiposity, which fail to meet the demand for new T-cell output [37,38]. Data resulting from the measurement of thymic emigrants by flow cytometer [39,40,41] or TCR excision circles (TRECs) by real-time PCR [37,42,43,44] have demonstrated that the release of new T cells decreases progressively during aging. The holes created in the lymphocyte compartment by low thymic output lead to a gradual increase in effector memory cells, thus old people have low numbers of naïve T cells and a high number of memory T lymphocytes, mainly cytotoxic CD8+ cells in an advanced stage of differentiation [13]. In addition, a restriction on TCR repertoire diversity [45], both in memory CD4+ and CD8+ T cells, occurs [46]. It is of interest that the accumulation of highly differentiated T cells and the significant reduction in naïve T cells are not observed in long-lived individuals [47], so centenarians show unexpectedly larger TCR repertoires [48].

The production of B lymphocytes, as well as of myeloid cells, is impaired in aged bone marrow [29]. Although the output of new B cells, quantified by measuring K deleting recombinant excision circles (KRECs), remains conserved [49], with the peripheral number unchanged, aging is associated with a decline in the frequency of naïve B cells and in the percentage of switch memory B cells [50,51]. Moreover, the accumulation of a subset of atypical B cells, termed age-associated B cells (ABCs), characterizes the aging B-cell compartments. These cells have distinct phenotypes, gene expression profiles, special survival requirements, variations in B-cell receptor repertoires, and unique functions [52]. As a result, antibody affinity and diversity also decline, leading to impaired antibody responses [53]. The exact cause and significance of all these changes are not clear, but alterations to immune cell signaling may be one of prominent cause of malfunctioning immunity, as extensively discussed in Fulop et al. [54].

Figure 2 summarizes the principal age-related modifications of the innate and adaptive immune system cells.

Aging affects several immune system tissues and organs. Increased architectural (as well as vascular) fibrosis induces a progressive reduction in lymph nodes number and size, leading to reduced local cell traffic and impaired intercellular interactions [55,56,57]. In addition, changes in the local production of adequate amounts of chemokines and cytokines were observed, which ensured an efficient immune response [57,58]. Aging also causes deterioration in the spleen. This organ preserves the physiologic populations of white blood cells and platelets that, mobilized by pathogenic invaders in the blood, sustain a protective immune response [59]. With advancing age, the sinusoidal stromal cell linings, at the border between the follicular and marginal zones, become disorganized, leading to altered immune system cell localization, resulting in an improper antigen-presenting capacity [60,61]. Finally, evidence has been gathered of an overall decline in mucosal immunity, especially in the gastrointestinal tracts of the elderly [62].

### 2.2. Sex-Differences in Innate and Adaptive Immunity Increase with Age

As sex differences characterize the immune system during the entire life course, further ones occur during aging, with older men displaying higher monocyte activity [63], expression of myeloid cell-related genes [64], and a faster decline in the number and function of B and T cells [63].

Aging female T cells produce more IL-10 [65], capable of neutralizing age-related inflammaging. The high humoral response observed in women can favor the appearance of autoreactive clones [66], as supported by epidemiological studies showing that about 80% of autoimmune diseases occur in females at earlier age [67]. Further, immunosenescence develops later and slower in women [68], and this has been associated with their longer life expectancy [69,70]. Unfortunately, this longer life expectancy is associated with worse health conditions [17].

Although detailed information is lacking regarding sex differences in immune performance, a number of advances have been made. A study, enrolling 500 males and 500 females aged between 20 and 70, tested eight lymphocyte subpopulations, 3 million single nucleotide polymorphisms, and 560 genes of the immune system before and after stimulation with a series of infectious agents and a superantigen. The data demonstrated that age preferentially influences CD8+ cells, while sex has a specific influence on CD4+ cells and macrophages, as well as multiple genes related to the immune system. In contrast, genetic factors are correlated with only a few changes in immune system genes, but these changes are extremely puissant [71]. Another study, performed with Transposase Accessible Chromatin with Sequencing (ATAC-seq), demonstrated that the switching on and off of some genes follows different individual patterns, with a random array of activation. In men and women, the turning on and silencing of immune system genes occur through different “switches”, which partially explains why the two sexes may differ in percentages and types of immune players [72]. Although both sexes equally respond to drugs targeting immunosenescence [68], women are reported to be more sensitive to immunostimulating strategies [73]. Understanding how sex differences in innate and adaptive immunity evolve with age is important for developing personalized approaches to health and vaccination strategies.

### 2.3. Immune System Complexity Increases with Age

Since only 30% of the variations observed in the immune system appear to be due to hereditary factors [74], a model has been proposed in which the immune response evolves over time in main waves shaped by environmental stimuli, leading to divergences in the immune system that become more and more marked as the years pass [75]. The first dramatic change occurs in the newborn, requiring protection from invasive pathogens and tolerance to “beneficial” pathogens. Cord blood cells show extreme individual variability, which flattens during the first weeks of life, including an initial short-lasting neutrophil expansion, and the increase in CD4+ and CD8+ lymphocytes and B cells occurring in all children [76]. B cells, NK cells and DCs reach adult-like maturation status in the first trimester, following increased exposure to a variety of environmental agents. The progressive senescence of the immune system during adulthood occurs earlier than was previously appreciated, most likely because comparative analyses were performed between young subjects and very old people, thus missing the intermediate age [74,77]. The limit of this comparison was bypassed by comparing homozygous twins with an average age of 48 years with those aged between 12 and 30 years. The study demonstrates that the inter-individual variability of most immune system subpopulations starts before the age of 50 [78], well before the age of 60, conventionally established as the initiation of immunosenescence.

One should remember that, to appreciate significant changes related to IS, long-term follow-up analyses should be performed, since a rather stable immune system cell composition is commonly observed in the short period of observation [79]. Indeed, a trend of naïve CD8+ cell expansion in the young and a decrease in the elderly was observed in a 10-year follow-up. However, this does not necessarily translate to the individual level: for example, naïve CD8+ T cells decline over time at the group level, but not individually, because in some cases, the variability over time proceeds in an opposite direction. This individual variability has been observed for other subpopulations, such as B cells, CD8+CD28− T cells, CD8+ T cells expressing the CD57 marker, and NK cells [77]. Current knowledge on human immune variability comes mainly from well-selected population-based cohort studies. Therefore, to promote a better understanding of the impact of immune variability on aging, these approaches must be extended to populations of different backgrounds, life stages, lifestyle, and environments.

### 2.4. Biomarkers of Aged Immune System

As described above, the immune system ages and the changes that occur can be highlighted by studying specific markers of aging. Converging literature recognizes IL-6 and TNF-α as relevant biomarkers for the aging innate immune system [80]. While IL-6 serum concentrations in healthy individuals are undetectable or minimal, elevated levels have been reported, especially in advanced age, to be associated with disability and mortality; TNF-α age-related changes behave similarly [81]. The production of other key cytokines important for immune modulation, including IL-7, IL-11, IL-15, and granulocyte-macrophage colony-stimulating factor, is also affected by aging [29,80]. Although total white blood cell counts and lymphocyte counts have been the most studied biomarkers of adaptive immunity, they can further differentiate after their maturation in response to pathogens, so it is unclear when senescence is induced in these cells [82]. Similarly, C-reactive protein plasma level, which is a frequently used screening test in daily clinical practice, seems to be a rather unspecific biomarker of the aging immune system.

Telomere length measure, recognized as one of the most reliable biomarkers of aging [83], provides a rough estimate of the rate of the immunosenescence process and can hardly be regarded as clinically relevant [84]. More informative is the quantification of TRECs and KRECs [43], which mirrors the capability to homeostatically replenish both lymphocyte pools. A decline in the diversity of the TCR repertoire owing to thymic involution has been implicated as causing defective immune responses in the elderly [46].

Upfront to a lack of immune cell global markers of senescence, individual immune system cell lineages have been identified. T cells of aged individuals express a CD27−CD28−CD57+ killer cell lectin-like receptor G1(KLRG-1)+ or C-C chemokine receptor 7 (CCR7)−CD45RA+ phenotype. These cells can also express T-cell immunoglobulin and mucin domain-containing 3 (Tim-3), T-cell immunoreceptor with Ig and ITIM domains (TIGIT), immunoglobulin-like transcript 2 (ILT2/CD85j), or other NK-like receptors. Unsettled is the issue of whether T cells really express exhausted markers such as programmed cell death protein 1 (PD-1) and lymphocyte-activation gene 3 (LAG-3) [85]. With aging, a progressive increase in terminal effector memory T (TEMRA) cells occurs [86]. These T cells, which are CCR7-, re-express the CD45RA marker and represent terminally differentiated effector cells that result from a protracted antigen exposure. Overall, TEMRA cells, characterized by a decline in proliferation potential, efficient cytotoxicity, pro-inflammatory activity, and high clonal expansion, are considered hallmarks of immunosenescence [87,88]. Finally, senescent T cells also become positive for senescence-associated β-galactosidase (saβ-gal) staining, upregulate p53, p21, and p16, downregulate cyclin-dependent kinase (Cdk) 2, Cdk6, and cyclin D3, and show SASP [85].

Since aging negatively affects the production of B cells in the bone marrow, the number of B-1 and B-2 cells is decreased in peripheral blood, counterbalanced by the increase in ABCs [52]. Although the phenotype of surface markers is still poorly defined, ABCs could be distinguished from other B-cell subsets because they express CD11b, CD11c, and T-bet markers, as well as innate activation stimuli, such as Toll-like receptor 7 signals, but not CD43 and CD5 molecules, which are present on B-1 cells [53].

Monitoring these biomarkers can provide insights into the aging immune system and help identify individuals who may be at increased risk for age-related immune dysfunctions and related health problems.

### 2.5. The Systems Immunology for the Study of Age-Related Immune System Variations

The “classic” study methods, resorting to the use of animal models, despite having highly contributed to unveiling relevant paradigms in immune system, have some imbedded limitations, such as genetic homogeneity, a synchronized day–night cycle, monotonous feeding, and exposure to a narrow spectrum of environmental stimuli. Therefore, animal modes lack those conditions that are so important for shaping the human immune repertoire [89]. Some investigations have used wild or pet store mice, which are maintained in hygienic conditions, which may recapitulate adult human immune challenges [90]. More recently, the issue of immune system variability has been addressed by a new approach, defined as “system immunology”, a branch of “systems biology” [91]. This methodology is based on the concept that various components of the immune system have a high degree of interdependence and interconnection. In a multihierarchical or multiscale organization, the components at a lower scale are integrated with the functional units of the immediately higher scale [92]. The variation in cell composition, plasma proteins, and functional responses in aging through a system immunology approach has been possible due to the introduction of high-throughput “omics” technologies [93]. This approach allows us to study, at both the single cell and population level, the behavior of genes and epigenetic modifications (genomics and epigenomics), mRNA (transcriptomics), and proteins (proteomics), and to quantify, in blood and tissues, the levels of immune system components, as well as the immune system cell markers, as summarized in Figure 3.

Advancements in mathematical and computational methods allow us to study the interactions within cellular and molecular immune system networks. This provides unprecedented information on the human immune response following vaccinations, infections, autoimmunity, and tumors [94]. Many of these improvements have been optimized for studies at the single cell level [95], especially to characterize immune system cells that are located in multiple sites, i.e., blood, primary and secondary lymphoid organs, respiratory and gastrointestinal tissues (lungs, intestine, pancreas, and liver), and barrier sites (skin and mucosal surfaces), where the composition, phenotype, function, and tissue-specific adaptations of different immune cell populations are known to differ [96]. One study that relies on use of these new technologies in addressing immunosenescence is the S3WP program, consisting of a cohort of 101 Swedish individuals between 50 and 65 years of age, followed longitudinally for 2 years. In this program, repeated immune cell profiles, proteomics, transcriptomics, lipidomics, metabolomics, and autoantibody data were integrated with information on gut microbiota composition, routine clinical chemistry, and various clinical parameters [97].

## 3. Aging of Immune System Components in Infectious Diseases

The interplay between immune senescence, inflammaging, and infections is complex and multifaceted, shaping the susceptibility, severity, and outcomes of infectious diseases in aging individuals. It has become a growing field of research aimed at clarifying whether pathogens accelerate the aging of immune system cells by inducing the expression of molecular determinants overlapping those associated with cellular aging, or whether the age-induced immune changes facilitate infection burden.

Infectious diseases, such as influenza and pneumonia, are among the main global infectious killers of the elderly, with a rate of 93.2 deaths per 100,000 in 2018 for those aged ≥65 years [98]. Moreover, infections by new pathogens, such as West Nile virus (WNV) and severe acute respiratory syndrome coronavirus 2 (SARS-CoV-2), responsible for the coronavirus disease 2019 (COVID-19) pandemic, have shown heightened severity in the older population [99,100]. Overall changes in non-immune organs also concur with the age-related proneness to infectious diseases. For instance, in the lungs, decreased respiratory muscle strength reduces lung compliance, and impaired muco-ciliary function leads to inefficient clearance of infectious organisms [101]. Similarly, the decreased function of the epithelial barriers of the skin, lungs, and gastrointestinal tract favors the pathogen invasiveness [102]. Furthermore, neurocognitive modifications and increased vulnerability to metabolic encephalopathy contribute to the delayed recognition of infectious syndromes [103]. This is further aggravated by the frequent atypical clinical presentation of infections in frail older adults, often identified at the time when the underlying disease worsens or when an additional precipitating event occurs [104].

However, it is now recognized that age-related changes in immunity play a major role in determining the incidence and severity of infections and influencing spontaneous- or vaccine-induced immunity. In addition, persistent infections result in a prolonged antigenic overload of the immune system, leading to host T-cell exhaustion [105] and the promotion and/or acceleration the progressive accumulation of cells with an immunosenescent phenotype [106], thus easing the development of a pro-inflammatory milieu that favors aging-related diseases [107,108]. Therefore, both chronic infection and the related immune response contribute to accelerating aging [109]. This acceleration is anticipated to be largely variable, being related to the variety of inducing environmental factors, type, intensity, duration, and temporal sequence of antigen exposure, the so called “exposome” [110], implying that that aging is an individual phenomenon. Indeed, while chronological age is strongly correlated with the prevalence and severity of infections, it may not be the only cause of the higher susceptibility. In addition, the probability of a serious infection does not increase on a yearly basis, but it is closely related to concomitant chronic diseases that induce a worsening of the inflammatory state. Therefore, an elderly but healthy person may be more protected from serious infections than a younger person in poor health [111].

While several studies have focused on chronic diseases in the elderly, less is known about the epidemiological characteristics of their infectious diseases. In a French survey performed in 2005, the top five infections included, with decreasing incidence: urinary tract infection (UTI), respiratory tract infection, skin and soft tissue infections, surgical site infections, and bacteremia [112]. Also, intra-abdominal infections, such as cholecystitis, diverticulitis, appendicitis, abscesses, infectious endocarditis, bacterial meningitis, tuberculosis, and herpes zoster, have been reported to frequently occur in the elderly [113]. More recently, the availability of electronic records and health data platforms outlined the epidemiological characteristics and changes in infectious diseases among the elderly in the Shandong province of China [114]. This study outlined that respiratory infections have the highest incidence in this population, followed by mucocutaneous, blood- and sex-transmitted, gastrointestinal, and vector-borne infections. The most frequent respiratory diseases were influenza, pneumonia, pulmonary tuberculosis, cryptococcosis, whooping cough, and aspergillosis. Typhoid was by far the most frequent gastrointestinal infectious disease, followed by other infectious diarrheas, paratyphoid, ascariasis, bacteria and amoebic dysentery. Herpes zoster, viral conjunctivitis trachoma, common warts, and herpes simplex infections were prevalent among mucocutaneous diseases, while hepatitis B, syphilis, gonorrhea, and hepatitis C were listed as the most frequent blood- and sex-transmitted infections [114]. This study also reported that the incidence of infections was prevalent in elderly males.

### 3.1. Viral Infections

Viral infections represent one of the most important drivers of premature aging.

Human immunodeficiency virus (HIV) infection has been the first described as associated with immunosenescence [115]. Premature aging is also a consequence of highly active antiretroviral therapy, which has prolonged the survival of these patients despite a persistent inflammation, fueled by high levels of IL-6 and TNF-α, interferon (IFN)-α, reduced production of IL-2 [116,117,118], as well as by a variable decrease in the number of TRECs [37,119,120]. Using an assay that simultaneously measures TRECs and KRECs [43,49], we investigated the bone marrow and thymic output in treated HIV-infected patients and in those ones not requiring therapy. Cells containing KRECs remained unchanged after one year of therapy, then decreased to levels that were similar to those of untreated HIV+ patients. TRECs production increases following a successful antiretroviral therapy response, but never reaches the levels of HIV-1-infected patients not requiring therapy [121]. During this infection, an expansion of TEMRA cells expressing exhaustion markers has been described, strongly suggesting the involvement of T-cell-mediated immune responses in the infection associated immunosenescence. The inflammatory microenvironment of these patients may also drive TEMRA cell evolution towards non-specific senescence [88]. Very recently, the level of immunosenescence was investigated based on CD27 and CD57 expression on T cells in subjects living with HIV/AIDS. These subjects showed a lower proportion of T cells in the early stages of senescence and a higher proportion of T cells in the intermediate and final stages of aging [122]. The lowering of thymic output and the increase in TEMRA cells can be responsible for the observed oligoclonal expansions in both CD4+ and CD8+ T-cell compartments and a reduction in TCR diversity [123,124].

NK cell senescence in chronic HIV-infected patients is documented by the expansion of non-functional CD3−CD56−CD16+ NK cells displaying an activated profile, as indicated by the higher levels of cytokines and chemokines production, such as IL-4, IL-5, IL-6, IL-10, IL-12, IFN-α2, IFN-γ, TNF-α, RANTES, and MCP-1 [125]. Therefore, during HIV infection, the immunosenescence process is accelerated, resulting not only in an immunosuppressed state unable to contain HIV replication and spread, but also to a lower capacity to respond to new antigenic challenges favoring other infections and age-associated organ diseases [126].

For a discussion of the role of aging in response to severe acute respiratory syndrome coronavirus 2 infection, we recommend the reader refer to recent contributions [118,127,128].

As anticipated, the elderly population is more frequently prone to infections of the respiratory system caused by adenovirus, coronaviruses, human metapneumovirus, influenza A and B, parainfluenza, and respiratory syncytial virus (RSV). While in many countries, the vaccination rate is over 60% in subjects older than 65 years [129], influenza remains a serious health threat to this population. Disability and mortality outcomes are dependent on the level of frailty, cardiovascular events that are the most common extra-pulmonary complications, and diminished quality of life due to loss of independence following hospitalization. Unfortunately, large gaps exist between our understanding of the immune response to a natural influenza infection and vaccination, although immunity appears to be more robust and longer-lived in natural infections than following vaccinations [130]. Undoubtedly, a failure of innate immunity is also likely to play a role because of the decrease in macrophage functions [131,132], and of reduced uptake of opsonized bacteria by CD14+ monocytes [133]. Aging appears to affect DCs, in terms of a decrease in number, activity, and migration, determining poorer viral clearance as well as clinical outcomes and susceptibility to complications, which have been correlated to the levels of circulating inflammatory cytokines [134]. Although a diminished CD8+ T-cell immune response to influenza virus infection [135] has been demonstrated in the past in mice, only very recently has the role of CD8+ T cells been elucidated as a mechanism of respiratory viral infection severity in the elderly [136]. Single-cell profiling analysis demonstrated that CD8+ T cells specific for the major influenza HLA-A*02:01-M158-66 (A2/M158) epitope are similar in infants, children, and old humans. However, although CD8+ T cells of old adults displayed no signs of exhaustion, their suboptimal TCRαβ signatures led to less proliferation, polyfunctionality, avidity and recognition of peptide mutants. In particular, the reduced public TCRαβ clonotypes and TCRαβ diversity within older TCR repertoires explains why older adults, in the absence of pre-existing antibodies, are at higher risk of severe disease during influenza epidemics and pandemics [137]. Elderly people often have recurrent clinically mild RSV infections [138], which occasionally may cause altered airway resistance and the worsening of chronic obstructive lung disease [139], leading to serious or life-threatening pneumonia [140]. This population has low levels of serum neutralizing antibodies and IFN-γ and diminished in vitro cellular immune responses to RSV [141]. Data from a recent multiparametric immunological analysis suggest a primary role of cellular immunity in preventing symptomatic RSV infections in these patients. The study identified certain RSV-specific effector memory T-cell populations that are associated with the prevention of symptomatic infection, thus representing a potential marker of innate immunity to RSV in older age [142]. In addition, a systematic literature review of the risk factors for poor outcomes in RSV infection found a strong link between immunosenescence and preexisting co-morbidities (cardiac, pulmonary, and immunocompromising conditions, diabetes, and renal disease), and living conditions (socioeconomic status and nursing home residence) [143]. Other respiratory viruses, such as human metapneumovirus, human rhinovirus, and human parainfluenza virus, also lead to substantial morbidity and mortality in the elderly [144], but the impact of senescence and other factors modulating their severity in the elderly remain to be identified.

Despite being extensively investigated, no conclusive data is available regarding how cytomegalovirus (CMV) infection influences the immune response in the elderly and how it can trigger an increase in pro-inflammatory cytokines, promoting inflammaging [145]. Most likely, CMV infection contributes to age-associated changes in adaptive immunity by modulating the frequency and the cytotoxic capacity of NK cells. CMV seropositive elderly donors have a decreased percentage of both CD16− and CD16+ CD56bright NK cells, and an increase in the CD56−CD16+ subset [146]. An exploratory study demonstrated that CMV-specific CD8+ T cells of the elderly display a high expression of CD244, the T-cell differentiation marker of effector cells, suggesting the role of these lymphocytes in age-associated defective immune responses [147]. Chronic antigenic stimulation induced by persistent CMV infection drives a state of peripheral T-cell compartment exhaustion in older individuals, who exhibited an absolute increase in the effector memory CD4+ and CD8+ cells [148]. This increase occurs prevalently in the presence of elevated anti-CMV antibody titers. It is of interest that, although the number of CD8+ naïve T cells is lower in the blood, whether or not they are CMV-seropositive, only CMV-infected older adults have a lower number of naïve CD4+ T cells [148]. The accumulation of terminally differentiated, apoptosis-resistant, CMV-specific CD8+ lymphocytes is believed to reduce the overall TCR repertoire diversity [149,150]. This defect may impair the ability of older adults to respond to antigens they have never previously encountered. Accordingly, recent RNAseq data provide circumstantial evidence that aged subjects, and CMV-seropositive individuals in particular, exhibit blunted responses to neoantigens, partially due to reductions in the numbers of naïve T cells, resulting in a reduced TCR repertoire diversity [151].

In healthy donors >60 years of age, coinfection with CMV and Epstein–Barr virus (EBV) drives expansion of CD56- NK cells, characterized by reduced cytotoxic capacity and IFN-γ production [152]. Antigen-specific T cells against EBV undergo an age-related increase in the expression of markers associated with a differentiated phenotype, including KLRG-1, an increase in terminally differentiated T cells, and a decrease in the TCR repertoire diversity [153]. In infected patients, the expansion of viral-specific, exhausted, senescent CD8+CD28− T cells seems to play a central role in the onset of neoplastic lymphoproliferation, although the pathophysiology varies enormously among different disease entities [154]. However, information regarding the global burden of immunosenescence in lymphomagenesis is still scant, and requires a detailed analysis of different types of lymphoproliferation.

Long after hepatitis C clearance, predominantly male patients had increased plasma levels of SASP proteins, including IL-1α, IL-1RA, IL-8, IL-13, and IL-18. These changes have been associated with an increased risk of developing liver and non-hepatic diseases [155]. Infected patients likely produce an increased number of intrahepatic senescent, not functional T cells [156,157], which may promote the development of hepatocellular cancer as these T cells would be unable to eliminate premalignant senescent hepatocytes [158].

Varicella zoster virus (VZV) induces an expansion of NK cells displaying the terminally differentiated senescent marker CD57 [159], inhibiting their ability to secrete cytokines and to lyse virally infected target cells through NK cell-dependent cytotoxicity. In addition, the virus can interfere with the type 1 IFN pathway and the production of pro-inflammatory cytokines [160]. The incidence of herpes zoster, due to VZV infection, is also associated with an age-related decline in cell-mediated immunity against the virus, namely with the reduced frequency of virus-specific effector memory T cells [161], which appears to be the cause of the increased risk of post-herpetic neuralgia complications [162]. This is an important issue in view of the expected increase in population aging, leading to a higher incidence of herpes zoster cases in developed countries [163].

Other examples of the relationship between viruses and immunosenescence are measles, parvoviruses, and dengue viruses, as well as Merkel cell polyomavirus infections. Infection of human lung fibroblast cells by the measles virus generates cellular senescence, as documented by reduced cell proliferation, saβ-gal activity, increased expression of p53 and p21, as well as the induction of pro-inflammatory secretome with IL-8 or C-C motif chemokine ligand 5 expression [164]. Similarly, parvoviruses induce the expression of a pro-inflammatory secretome, including IL-8, IL-6, and IL-1β production [165]; dengue virus and Merkel cell polyomavirus elicit saβ-gal expression [166,167]. Whether these virus-induced senescence-mediated pathways is presently unknown.

Several signs of immunosenescence have been observed in the viral encephalitis caused by WNV [168], representing a worldwide health concern due to the prevalence, severity, and lack of efficient treatments for this disease. This virus is recognized by pathogen recognition receptors, the Toll-like receptors, which undergo an age-related decline [169]. Also, neutrophils, monocyte/macrophages, and DCs, the first barriers to this infection, show an age-related impairment [99]. In particular, an age-related up-regulation of AXL receptor tyrosine kinase, a molecule that, by regulating the blood–brain barrier permeability, facilitates viral uptake through phospholipid binding, has been reported. This receptor, potentially relevant for susceptibility to WNV, has been found in human DCs [170]. NK cells show age-related changes in phenotype and function [171].

Finally, it must be remembered that impairments of the immune system resembling those induced by aging can have an iatrogenic origin; this is the case, for example, of disease-modifying therapies for multiple sclerosis [172]. CD4+ T-cell lymphocytopenia, particularly within the central nervous system, low production of TRECs and KRECs, and T-cell repertoire restrictions increase the susceptibility of patients to infections, as reported for patients treated with natalizumab, who suffer of an earlier onset of progressive multifocal leukoencephalopathy, a rare but potentially fatal opportunistic infection caused by the JC virus [172,173,174].

### 3.2. Bacterial Infections

Several bacterial infections have been described as associated with the development of immunosenescence. This is the case with *Mycobacterium tuberculosis* (MTB). Although most individuals exposed to MTB manage to control tuberculosis (TB), which can remain in a latent form, approximately 5–10% of them develop an active disease [175]. Older adults are particularly susceptible to recurrences of dormant infections due to delayed diagnosis, co-morbidity, increased institutionalization, and immunosenescence [176]. Indeed, as with many other aging-related diseases, the diagnosis of active TB is difficult because of non-specific symptoms, such as unexplained fever and weight loss, which are less pronounced in this population [177]. In addition, age-related immunosenescence may reduce the overall sensitivity of traditional tuberculin skin tests and IFN-γ-releasing assays, which are recommended by the World Health Organization; this gap may be bypassed by the use of newer generation IFN-γ-releasing assays that show better performance in detecting TB infection among older adults [178,179,180]. Comorbidities, such as diabetes, could cause a ≥1.5-fold increased risk of developing TB [176]. The interactions between MTB and the host immune system are complex and only partially clarified, but the concomitant overall senescence of both innate and adaptive immunity appears to play a major role [181,182,183]. Changes in pulmonary IL-2 and TNF production may possibly impact granuloma formation and the maintenance of chronic infection. Similarly, an imbalanced pro- and anti-inflammatory factor pattern during aging may have a significant influence on the pulmonary macrophage functions relevant for the fusion of the phagosome with the lysosome, instrumental for eliminating MTB [184]. These data are in line with those obtained in mice, indicating that age induces an inflammatory pulmonary environment, turning the resident macrophages more prone to being infected by MTB [185]. Alterations in monocyte percentage and phenotype increase with age, suggesting a potential link with the elderly’s specific susceptibility to developing active TB [186]. Among older patients with TB, an age-associated over-representation of regulatory T cells, along with a significant reduction in the IFN-γ/IL-4 ratio, has also been reported, along with a significant reduction in other pro-inflammatory effector T-cell cytokines, such as IL-17A, IL-2, TNF-α, and polyfunctional (IFN-γ+ TNF-α+) T cells. This underscores the defective production of these effector cytokines, which may eventually lead to immunosuppression among older TB patients [187]. The studies demonstrating impaired adaptive T-cell immunity in old individuals with latent or active TB led to conflicting results, which do not allow us to identify immune age-associated changes as having a major pathogenic role [118,186]. Finally, it should be underlined that, despite the overall tuberculosis burden and rate of mortality in the elderly having declined in recent years [188], this pattern is expected to reverse, particularly for those aged ≥80 years [189]. Overall, due the low adherence to treatment and poorer tolerance to TB drugs, anticipatory diagnosis in the elderly represents an important challenge [176].

An additional leading cause of infection-related mortality in the elderly is *Streptococcus pneumoniae* infection [190]. In these patients, both antibody opsonic activities for all tested pneumococcal serotypes and phagocytic killing of pneumococci by neutrophils were significantly impaired [191]. The increased senescence markers, such as IL-1α/β, TNF-α, IL-6, and C-X-C motif chemokine ligand (CXCL)1, and cellular senescence facilitate bacterial adhesion to cells in the lungs [192], and may compromise upper respiratory mucosal immunity [193]. In addition, CD27+IgM+ B cells, which provide protection from *Streptococcus pneumoniae* and show age-related changes, may account for a reduced response to bacterial antigens [194].

Infections from *Escherichia coli* are the most common cause of recurring UTIs that continue to be a burden on aging females [195]. Older women are also more likely to be infected with the less common Gram-negative bacteria, such as *Klebsiella pneumoniae*, *Proteus mirabilis*, Enterobacter species, Citrobacter species, and *Pseudomonas aeruginosa*, as well as Gram-positive bacteria, including group B *Streptococcus*, other streptococci, *Staphylococcus aureus*, coagulase-negative staphylococci, and Enterococcus species [195,196]. The urinary bladder has physical barriers that prevent these infections, together with CD14+ monocyte-derived mononuclear phagocytes, and T cells [197]. The understanding of how immune senescence is related to UTI remains a challenge, although the data obtained in mice indicate a prevalent immune pathogenesis. It is of interest that, while T-cell responses can contribute to antigen-specific immunity in UTIs, bladder macrophages that in aged bladders begin to express CXCL13, may be responsible for inhibiting the development of adaptive immune responses [1,198]. In addition, macrophage phagocytic efficiency mediated via interaction with the NF-κB pathway during *Pseudomonas aeruginosa* infection has been found to be decreased [199]. The aged bladder produces high levels of the pro-inflammatory cytokines IL-6, IL-1β, and TNF-α [195], and is characterized by structures termed “tertiary lymphoid tissues” showing a germinal center with B-cell and T-cell zones and a follicular DC network. In addition, a redistribution of the B-cell pool from the periphery to the mucosal surface that alters the mucosal landscape has been described [195]. Interestingly, abundant γδ+ T cells were identified in the bladder [200], and experiments in mice demonstrated that these cells rapidly produce IL-17 upon infection, which promotes bacterial clearance [201]. Overall, the median percentage of γδ+ T cells decreases with increasing age [202].

Also, the overactive bladder, common in old women, appears to be correlated with inflammaging because the levels of nerve growth factor, C-C Motif Chemokine Ligand 2, and CXCL1 chemokines in the urine increase with age and disease severity [195].

Poor intestinal mucosal immunity represents a major factor leading to higher mortality from infections in aging [203] because it favors the changes in the human intestinal microbiota associated with inflammatory bowel diseases, irritable bowel syndrome, and metabolic disorders [204]. The altered balance between Gram-positive and Gram-negative intestinal bacteria may lead to the activation of DCs within the lamina propria of the intestine. As gut biodiversity decreases, potential pro-inflammatory microbes accumulate [205]. This imbalance starts a cascade of events inducing the release of pro-inflammatory cytokines, mainly IL-6 and IL-17. This, along with a decreased secretion of mucus and α-defensins by intestinal epithelial cells, further favors the entry of pathogens into mucosal layers [206]. Age-related alteration of the microbiota also leads to a decrease in the production of short-chain fatty acids, which may promote inflammation and cell vulnerability [207]. Dysregulation of the gut microbiota, more precisely the decrease in diversity of the microbiome associated with aging, appears also play a role in *Clostridium difficile* colitis, the major cause of gastrointestinal infections worldwide, occurring in up to 80% of cases in adults aged 65 [208,209]. Interestingly, patients who successfully eradicated *Helicobacter pylori*, another infection related to changes in the microbiota, had significantly lower histological markers of inflammation [210,211]. Finally, alteration in gut immune function has been proposed as one of the causes of small intestinal bacterial overgrowth (which implies excessive presence of bacteria) above 10^5^–10^6^ organisms/mL in small bowel aspirate [212]. This disease is common in the elderly, and is associated with chronic diarrhea, malabsorption, weight loss, and secondary nutritional deficiencies [213].

The Gram-negative anaerobic bacterium *Porphyromonas gingivalis,* responsible for a higher prevalence of periodontitis in the elderly population, is another pathogen associated with elevated expression of senescent cellular markers in immune cells, such as DCs. Indeed, bacterial invasion leads to an increase in the secretion of inflammatory exosomes which, in turn, amplify immune senescence [214].

Severe infections in the elderly can evolve into sepsis, a life-threatening event caused by host response failure, resulting in multiorgan collapse [215]. The outcomes of sepsis are worst in older adults, with higher rates of mortality, organ dysfunction, cognitive deficiency, and permanent disabilities [216]. These features can only partly be explained by age-related comorbidities [217], which are related to multiple organ failure produced by an excessive inflammatory response [218]. Bacteria highly responsible for sepsis are the Gram-positive *Streptococcus pneumoniae*, *Streptococcus progenies*, *Streptococcus agalactiae,* and *Staphylococcus aureus*; and the Gram-negative *Neisseria meningitides*, enteric (*Escherichia coli*, *Klebsiella*, *Proteus*, *Enterobacter*, *Serratia*, *Citrobacter*, and *Salmonella*), and non-enteric (*Pseudomonas aeruginosa* and *Acinetobacter*) [219].

The occurrence of sepsis arising from a concurrent excess of inflammation and immunosuppression [220] may also induce solid organ impairment [221], as in the case of blood–brain barrier disruption, local myocardial ischemia, or infarction secondary to preexisting coronaropathy. Liver damage appears in the early stages of sepsis in cirrhosis patients, in whom myeloid-derived suppressor cells are expanded. These cells, which inhibit the functions of DCs and macrophages [222], at the same time as reducing the diversity of NK cells, inhibit the Th1 response and induce Th2 and regulatory T-cell production [223]. Th2 cells have been shown to increase with aging [224], playing a paradoxical role in sepsis, since they may also increase the production of pro-inflammatory cytokines during emergency myelogenesis, and concomitantly be potently immunosuppressive [225]. Conclusively, our understanding of the relationship between aging and the onset of sepsis is incomplete and conflictual, as little or no association between age and inflammatory markers [226,227] or higher levels of these mediators among older septic patients [228] have been found.

### 3.3. Parasitic Infections

How age affects immune responses to lifelong parasitic infections is presently poorly understood. A supervised statistical learning technique indicated that older mice harbor a higher parasitic load than younger ones, due to the aging of adaptive immunity, characterized by reduced numbers of naïve T cells, poor T-cell responsiveness, and impaired antibody production [229]. In human cutaneous leishmaniasis, senescent CD56+CD57+ NK cells, together with CD4+ and effector memory CD8+ cells that re-express the CD45RA marker, play a role in the establishment and maintenance of tissue inflammation and are linked to lesion size [230,231]. RNA-seq data demonstrated that the cutaneous lesions exhibited a strong transcriptional co-induction of senescence-associated genes, and that the pro-inflammatory immune response is more strongly associated with the induction of senescent effector memory T cells re-expressing CD45RA [232]. In Chagas disease, a tropical disease representing a public health problem in developing countries, the infection by *Trypanosoma cruzi* causes excessive immune system stimulation that might elicit a progressive loss and collapse of immune functions. The induction of immune cells with senescent phenotypes may compromise the host’s capacity to control the magnitude of induced inflammation, predisposing infected hosts prematurely to immunosenescence [233], as demonstrated for CD4+ T cells. Indeed, in this infection, an increase in Ag-experienced IFN-γ-producing CD4+ T cells has been described [234]. Finally, there are data suggesting that the relationship among helminth parasite infection, immunity, and survival is not driven by genetics or early life environmental conditions, but rather by individual variations occurring late in life and, therefore, linked to immunosenescence. This was suggested mainly by a murine model performed in mice, in which Th2 function and anti-worm antibody production are compromised in old age [235].

Table 1 summarizes the principal age-related modifications of the innate and adaptive immune system occurring in age-related infectious diseases.

## 4. Immunosenescence and Vaccines

The interplay between immune senescence, inflammation, and infections highlights the importance of understanding age-related changes in the immune system and developing strategies to improve immune function in older individuals. The latter include the production of age-targeted vaccines. Vaccination that elicits immunological memory capable of protecting against subsequent infections is crucial in safeguarding the elderly, a population generally less responsive to both primary and booster challenges [134,236], and thus is less protected [237]. This was especially poignant in the wake of the COVID-19 pandemic, when the mortality rate was disproportionately high among the elderly [238]. The Advisory Committee on Immunization Practices of Centers for Disease Control and Prevention recommends that adults over 65 years obtain immunization against herpes zoster virus, *Streptococcus pneumoniae*, influenza viruses [104], and more recently, RSV in those over 60 years [239], and SARS-CoV-2 [104]. Moreover, tetanus and diphtheria vaccines were also recommended, despite responses to these vaccines often being impaired in older individuals [240].

The response to a vaccine is linked to several factors, such as the type, dose, and route of administration; the lack of, or incomplete, primary immunization; and the absence of regular boosters. Environmental factors, including exposure to pollutants and toxins, geographical location, seasons, the number of family members, composition of the microbiota, presence of co-infections, use of antibiotics, smoking, dietary intake, alcohol consumption, exercise, sleep patterns, and comorbidities of the host also play a role in vaccine response [241]. In addition, in older individuals, the response to vaccination is strongly influenced by immunosenescence and inflammaging. This last condition, through macrophage activation, creates a detrimental environment for the generation of a protective immune response to vaccinations, which is aggravated by the decreased antigen presentation capability of DCs to T cells [242]. Additionally, the production of cytokines is not optimal for adaptive immune response priming [243]. These responses in older individuals are also linked to the loss or decrease of the fine balance between the generation of inflammatory effector T cells and follicular helper T cells that mediate high-affinity antibody production, as well as the induction of long-lived memory cells for effective recall immunity [244]. In fact, during aging, this balance is reversed, and short-lived effector T-cell responses are prevalent over those of memory or follicular helper T cells [245]. Finally, vaccine-induced antibodies commonly show lower protective capacity [244], as suggested by the low production of antibodies in response to influenza vaccination in old rhesus macaques [246] and old individuals [247]. Similarly, IgA and IgM levels were found to be significantly lower in the elderly vaccinated against pneumonia [248,249,250], and antibodies against VZV are low in subjects ≥70 years old [251].

In view of the above, the currently available vaccines appear to provide short-term protection [241]. Therefore, new methodologies are needed to produce vaccines designed to optimally balance between immune stimulation and inflammatory status, and that are capable of inducing long-term immunological protection. These approaches include novel vaccine and adjuvant formulations, repeated heterologous booster injections, and alternative routes of administration [252]. New paradigms for the vaccine discovery and development process are based on “systems vaccinology,” which uses systems immunology technologies to probe the molecular networks that drive the immune response to vaccines. These advances, which incorporate immunobiography information with clinical, immunological and omics data, could allow stratifying subgroups of subjects and to identify those markers that could lead to the rational development of vaccines specifically designed for older adults [253,254]. The strengths of system vaccinology use are the possibility of exploiting results from previous studies on aging, in which the fixed variable is chronological age, with multiple longitudinal data, obtained from omics technologies, in which individuals are followed for a long time.

## 5. Limitations of Current Research on Immunosenescence, Future Research Directions and Potential Therapeutic Interventions

Although significant progress has been made in our understanding of immune features observed in the elderly in recent years, there are several common limitations in the studies regarding immunosenescence in both good health and diseases that should be considered. First, immune measures have usually been performed at a single point in time and, rarely, longitudinally. Therefore, there is still little information regarding the life-course trajectories of immune biomarkers or the differential rate of immunosenescence. Second, immunosenescence studies often lack appropriate controls. Third, characterizing cell phenotype in large-scale cohort studies is logistically and economically challenging, so the number of old subjects analyzed in different studies is usually limited. Fourth, a variety of particularities can arise in the context of the laboratory diagnostics of elderly people [255], and, therefore, it is possible to introduce technical errors due to transportation, pre-analytical mishandling, cell loss, and time involved to process samples. In particular, there are cell markers that are sensitive to the conditions in which they are collected, temperature, delay in processing, preservation method, duration of storage, and number of freeze–thaw cycles [256]. Fifth, although extensively studied, the knowledge of the exact molecular mechanisms related to immunosenescence remains limited, and molecular biomarkers for senescence remain lacking [15]. Sixth, there may be non-technical interferences, such as the frequent presence of acute diseases [257], which can momentarily alter immune features of old people. Finally, most studies have not considered sociodemographic differences in the aging population, while recent studies have demonstrated that individuals in more disadvantaged social positions experience greater levels of immunosenescence [258].

This review may also have some limitations. Because the literature related to immunosenescence has increased significantly in recent years and is continually updated, some information may have been lost, as well as information written in languages other than English.

Despite these limitations, the biases linked to individual variability and the conflicting results existing in the literature, the targeted and effective use of the large amount of information available today is the basis for the development of strategies aimed at fighting immunosenescence. At present, no preventive strategies capable of stemming the senescence of the immune system have been found. This is primarily because it is not yet well understood whether the observed immune function decline is a cause or a consequence of the interaction between the various systems within the aging of the organism. However, the identification of specific markers of cellular senescence will be crucial in the development of senolytic drugs targeting immune components. Several senolytic clinical trials have been completed, are current, or are planned for the near future [259], which include those focusing on immunosenescence or inflammaging [260,261,262]. While the first-generation senolytics demonstrated moderate efficacy when used in clinical trials, next-generation senolytics (CAR-T cells, antibody drug conjugates or vaccines) targeting specific proteins selectively expressed during senescence and, based on new technical developments, can pave the way for preclinical research [263]. Currently, the preclinical development of these strategies is challenging and aimed at avoiding strong side effects; however, the expected results are commensurate with the patients’ hopes regarding the treatments. Furthermore, stratification of the elderly population based on multiple biomarkers identified with the multi-omic approaches of system immunology, and the application of these techniques in the context of the mitochondrial genome [264] could help to recognize those patients who would benefit more from specific treatments, thus allowing more personalized approaches.

To our knowledge, no specific studies have evaluated the costs and burden of disease related to immunosenescence. However, it is important to underline that the diagnosis and treatment of chronic diseases related to age place a significant burden on National Healthcare System budgets [265]. Therefore, all strategies that will help to reduce immunosenescence, including the development of senolytics and successful vaccinations, have the potential to save many lives and to substantially reduce health care costs.

While current research on immunosenescence has provided valuable insights into age-related changes in the immune system, addressing existing limitations and pursuing innovative research directions are essential for developing effective therapeutic interventions to promote healthy aging and improve immune function in older adults. Collaboration across disciplines and integration of cutting-edge technologies will be key to advancing our understanding of immunosenescence and translating research findings into clinical practice.

## 6. Conclusions

The aged population worldwide is progressively increasing. According to the United Nations, the proportion of people over 65 will almost double between 2019 and 2050; therefore, age-related changes in the immune system are of enormous interest in the scientific and healthcare fields.

A large body of information has been accumulated regarding the still ill-defined link between bacterial infections and immunosenescence [266], indicating that aging affects the functions of immune system cells, resulting in an increase in infection severity, and that chronic viral infections can induce the senescence of immune cells.

Furthermore, it must be remembered that some younger individuals show features consistent with a pro-inflammatory and early immunosenescence state, which may predispose to an increased risk/severity of diseases [267]. On the contrary, there are elderly people, in particular centenarians, who harbor a unique, highly functional immune system that has successfully adapted to a history of insults, allowing for exceptional longevity [268]. These conditions can be related to the incapacity, at any age, to preserve and/or restore an optimal “immune resilience” when inflammatory or antigenic stressors occur [269]. Therefore, immune resilience would be a trait distinct from the processes that anchor immune status to chronological or biological age. Some individuals, especially males, may lack the ability to preserve optimal immune resilience when subjected to common inflammatory insults such as symptomatic viral infections.

Monitoring small populations of peripheral blood immune cells from clinical patient samples with new system immunology tools have enabled studies of cellular exhaustion leading to senescence directly in humans, especially during chronic infections. Despite this, to date, there are no precise data on the existence of common connections between immune senescence and specific infectious diseases. Mouse models have also been particularly informative in studies of aging [270], but they have not allowed us to establish to what extent these models faithfully recapitulate the mechanisms underlying immunosenescence processes in humans and their possible common effects in predisposing to infections. Similarly, clinical trials targeting aging in humans have shown promising but limited results [271]. In addition, all these studies did not consider potential changes in senescent cells residing in the target tissues of infections. A combination of multiple scientific approaches (e.g., multiparametric analyses, high-resolution omics technologies, systems biology and big data analytics) would enable a better understanding of common immune senescence features during human infections.

Although the full extent of the biological changes is largely unknown, several characteristic changes are typically and constantly observed in aging. Identifying hallmarks and characteristics associated with immunosenescence will be crucial for exploring their impact and significance, particularly in age-related infectious diseases.

## Figures and Tables

**Figure 1 microorganisms-12-00775-f001:**
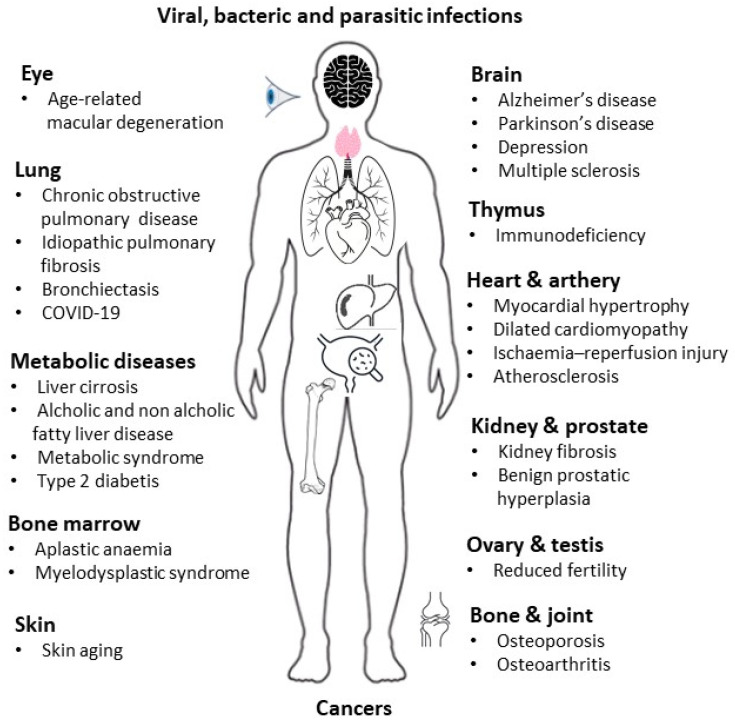
Schematic representation of the principal age-related diseases.

**Figure 2 microorganisms-12-00775-f002:**
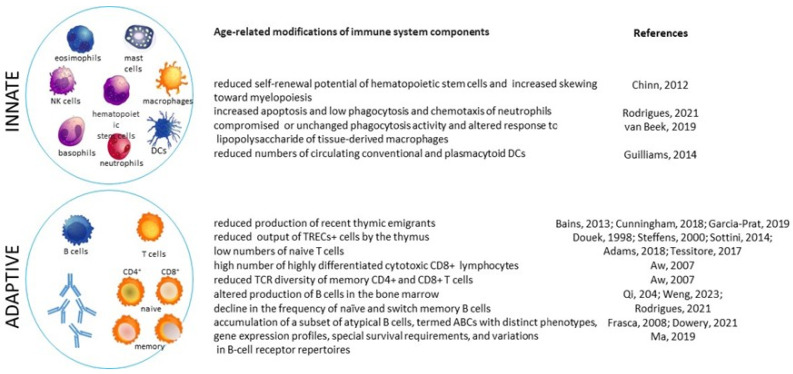
Main qualitative age-associated changes in the cells of innate and adaptive immunity [13,28,29,32,35,37,39,40,41,42,43,44,45,46,49,50,51].

**Figure 3 microorganisms-12-00775-f003:**
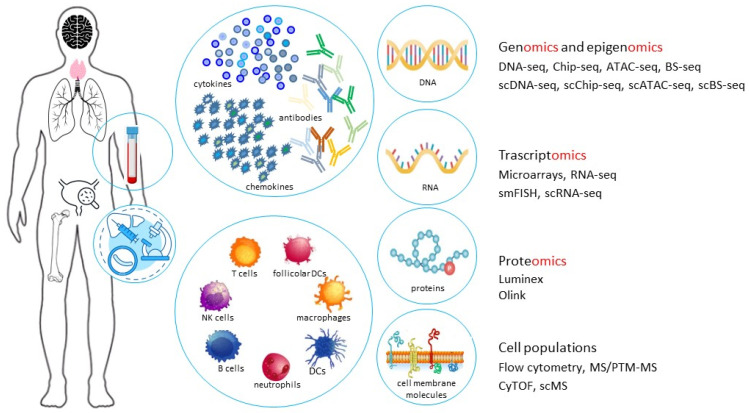
Overview of the principal organs, immune components, and targets of omics profiling technologies that can be used to characterize the age-related modification of the immune system at population and single-cell levels. ATAC-seq: assay for transposase-accessible chromatin using sequencing; BS-Seq: bisulfite sequencing; ChiP-seq chromatin immunoprecipitation assays combined with DNA sequencing; scMS: single-cell metabolomics; scRNA-seq: single-cell RNA sequencing; cyTOF: cytometry by time of flight; smFISH: single-molecule fluorescence in situ hybridization; and MS-PTM-MS: tandem mass spectrometry (MS/MS) of posttranslational modifications (PTM).

**Table 1 microorganisms-12-00775-t001:** Main age-associated immune defects observed in the principal infectious diseases affecting old individuals.

Infective Agents		Innate Immunity		Adaptive Immunity	
			References		References
viral	HIV	IL-6 and TNF-α elevation	[116,118]	decreased thymic output	[37,119,120]
		IFN-α increase and IL-2 reduction		expansion of TEMRA cells	[88]
		expansion of non-functional CD3-CD56-CD16+ NK cells	[125]	elevated KRECs in patients not needing therapy	[121]
				decreased T-cell diversity	[123]
				diminished CD8+ T-cell response in aged mice	[135]
				reduced public CD8+ TCRαβ clonotypes and TCRαβ diversity	[137]
	Influenza virus	decrease in macrophage peritoneal phagocytic function	[131,132]		
		reduced uptake of bacteria by monocytes	[133]		
		decreased abundance, activity and migration of DCs	[134]		
	RSV	low levels of serum neutralizing antibody and IFN-γ	[141]	RSV-specific effector memory T cells prevent symptomatic infection	[142]
	CMV			increased CD8+ CD244+ effector T cells	[147]
		decreased CD16-/CD16+CD56bright and increase in CD56−CD16+ NK cells percentage	[146]	lower number of naïve CD4+ and increased effector memory CD4+ and CD8+ T cells	[148]
				reduced TCR repertoire diversity	[151]
	EBV	expansion of CD56- NK cells with reduced cytotoxic capacity and IFN-γ production	[152]	increase in terminally differentiated T cells and decrease in TCR repertoire diversity	[153]
				expansion of viral-specific exhausted, senescent CD8+ CD28− T cells	[154]
	HCV	high plasma levels of SASP proteins	[155]	increase in intrahepatic senescent, not functional T cells	[156,157]
	HZV	interferes with the type 1 IFN pathway and the production of pro-inflammatory cytokines	[160]	reduced frequency of virus-specific effector memory T cells	[161]
		increases in CD57+ NK cells	[159]		
	Measles virus and parvovirus	induction of pro-inflammatory secretome-related factors	[164]		
	WWNV	impairment of neutrophils, monocyte/macrophages, DCs, and NK cells	[169]		
	JCV			CD4+ T-cell lymphocytopenia, low production of TRECs and KRECs and TCR repertoire restrictions in natalizumab-treated patients	[172,173,174]
bacterial	*Mycobacterium tuberculosis*	alterations in monocyte proportion and phenotype	[186]	impaired adaptive T-cell immunity	[118,186]
		reduction in IFN-γ/IL-4 ratio and other pro-inflammatory, such as IL-17A, IL-2, TNF-α	[187]	reduction in regulatory T cells and polyfunctional IFN-γ+TNF-α+ T cells	[187]
		imbalanced pro- and anti-inflammatory factor pattern and changes in IL-2 and TNF-α production in the lung	[184]		
	*Streptococcus pneumoniae*	low opsonic activities of antibodies and phagocytic killing of neutrophils	[191]	changes in CD27+IgM+ B cells	[194]
		increase in senescence markers (IL-1α/β, TNF-α, IL-6, and CXCL1)	[192]		
	*Escherichia coli* and other bacteria inducing urinary tract infections	high levels of pro-inflammatory cytokines (IL-6, IL-1β, and TNF-α)	[195]	formation of bladder tertiary lymphoid tissues and redistribution of B-cell pools from the periphery to mucosal surface that alter the mucosal landscape	[195]
		aged bladder CXCL13+ macrophages may be responsible for inhibiting development of the adaptive immune responses	[198]		
		decreased macrophage phagocytosis	[199]		
	Gram-positive and Gram-negative intestinal bacteria disequilibrium	activation of DCs	[205]		
		release of pro-inflammatory cytokines, mainly IL-6 and IL-17	[206]		
	*Porphyromonas gingivalis*	senescent cellular markers in DCs	[214]		
	Gram-positive and Gram-negative induced sepsis	expansion of myeloid-derived suppressor cells, inhibiting the function of DCs and macrophages in cirrhosis patients	[222]	inhibition of Th1 response and induction of Th2 and regulatory T-cell productions	[222]
parasitic	*Leishmania*	expansion of senescent CD56+ CD57+ NK cells	[231]	expansion of CD57+ CD4+ lymphocytes	[230]
				expansion of effector memory CD8+ T cells that re-express CD45RA marker	[231]
				increased transcriptions of senescence-associated genes in the cutaneous lesions	[232]
	*Trypanosoma cruzi*	compromised capacity to control the magnitude of inflammation	[233]	increase in antigen-experienced IFN-γ-producing CD4+ T cells	[234]
	Helminths			compromised Th2 function in mice	[235]

HIV: human immunodeficiency virus; RSV: respiratory syncytial virus; CMV: cytomegalovirus; EBV: Epstein–Barr virus; HCV: hepatitis C virus; HZV: herpes zoster virus; WNV: West Nile virus; JCV: JC virus.

## Data Availability

No new data were created.

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
