# Peer review of "The Impact of Immune System Aging on Infectious Diseases"

_microorganisms, 2024, doi:10.3390/microorganisms12040775_

Round 1
Reviewer 1 Report
Comments and Suggestions for Authors
The paper titled "The Impact of the Immune System Aging on Infectious Diseases" provides a comprehensive overview of the interplay between immune senescence and infectious diseases, particularly focusing on viral and bacterial infections in the elderly population. The paper covers a broad range of topics related to immunosenescence, aging-related changes in the immune system, and their implications. Although the paper provides a valuable contribution to the understanding of immunosenescence and its implications for aging-related diseases, there are some concerns regarding the data's validity and overall results.
1. The paper lacks detailed mechanistic explanations of how immunosenescence contributes to immune dysfunction and aging-related diseases.
2. Add specific examples or case studies that would enhance understanding.
3. Including schematic diagrams or flowcharts to illustrate key pathways or mechanisms involved in immunosenescence is recommended.
4. Addressing the limitations of current research, such as confounding factors, and discussing future research directions and potential therapeutic interventions would provide a more comprehensive overview.
5. A more in-depth analysis and interpretation of the data presented, along with a critical analysis of existing literature, including strengths, limitations, and gaps in knowledge, are necessary.
6. Findings from different studies should be integrated to provide a cohesive narrative.
7. Connections between immune senescence and specific infectious diseases should be drawn to highlight common mechanisms.
8. Each section should conclude with recommendations for future research directions, identifying unanswered questions, and suggesting potential study designs or methodologies.
9. Insights into recent advancements in vaccine formulations and adjuvants for aging populations should be included.
10. Discussions on the implications of immunosenescence for clinical practice and public health policies are needed.
11. A more in-depth discussion on the potential of senolytic drugs and multi-omic approaches in addressing immunosenescence, including examples of promising preclinical or clinical studies, is needed.
12. Minor points include enhancing the clarity and readability of the paper's English quality:
- Some sentences are quite long and complex, which may make them difficult to follow. Breaking them down into shorter, more concise sentences could improve readability.
- In some instances, there may be opportunities to use more precise or concise language. For example, replacing phrases like "The response to a vaccine is linked to several factors" with "Vaccine response depends on various factors" could make the writing more concise.
- The flow between paragraphs could be improved with more transition words and phrases. This would help guide the reader from one idea to the next more smoothly.
Comments on the Quality of English LanguageSome minor editing of the English language is necessary.
Reviewer 2 Report
Comments and Suggestions for Authors
This review article by Quiros-Roldan acknowledges the complexity of the immune system's changes with age, involving both innate and adaptive components, as well as the influence of various factors such as endogenous and exogenous factors and co-morbidities. The language used is technical but accessible, suitable for an audience familiar with immunology and public health concepts. However, it could benefit from a smoother flow in some areas for enhanced readability.
I have only some minor suggestions,
1. On page 2, lines 57-58, the statement about the senescence process should not be limited to only three cell types. As an orthopedic surgeon, I've observed that chondrocytes and tenocytes can also undergo senescence in patients with conditions like osteoarthritis and tendinopathy as they age. Please revise this sentence to encompass a broader range of cell types affected by senescence.
2. I recommend avoiding the abbreviation "IS" for the term "immune system" throughout the manuscript, as it may confuse readers. Please revert it back to "immune system" for clarity.
3. On page 3, lines 125 and 128, please use the correct spelling "naïve" instead of "naive".
4. To maintain consistency, I propose changing "NATURE" to "INNATE" in Figure 1 to more accurately describe the innate aspect of the immune system.
5. It would be beneficial for the authors to include a brief paragraph discussing NETosis in neutrophils or pyroptosis in macrophages during aging, as these emerging terms are important in understanding the behavior of these immune cell lineages.
6. I suggest that Figure 2 may be too simplistic for readers. The authors could enhance it by highlighting significant biomarkers such as IL-6, TNF-, p53, p21, p16, CD27−CD28−CD57+killer cell lectin-like receptor 228 G1(KLRG-1)+, or CCR7−CD45RA+, CD11b, CD11c, and T-bet markers in ABCs, and so forth.
7. In Table 1, it would be sensible to align "Natural" with "Acquired". However, in Figure 1, it's appropriate for "Innate" to be paired with "Adaptive" for clarity and consistency.
Reviewer 3 Report
Comments and Suggestions for Authors
This review summarises the main hallmarks of immune system aging, and their impact on immune answers in the context of infectious diseases and vaccination. The paper is well written, but does not cite the most recent updates in the field. Moreover it lacks a focus on molecular mechanisms involved, does not even mention the role of SASP, and exhibits no aspects/contents adding a unique value in comparison with other similar recent reviews in the field.
Comments on the Quality of English LanguageEnglish is fine enough, only a modest revision is recommended.
Round 2
Reviewer 1 Report
Comments and Suggestions for Authors
The paper has been significantly improved and is now much clearer. I believe that the authors have addressed enough comments to meet the acceptance criteria for publication in Microorganisms.
Reviewer 3 Report
Comments and Suggestions for Authors
This reviewer would like to thank the Authors for their reply. It is worth noting that in the attached file the Authors repeatedly mention one of their previous papers about the same topic. This appears as a confirmation that the topic was adequately discussed in the mentioned previous paper, and this under-review manuscript does not add any detail worth of a dedicated brand new paper.
Comments on the Quality of English LanguageEnglish is fine